# Molecular Action Mechanism of Coixol from Soft-Shelled Adlay on Tyrosinase: The Future of Cosmetics

**DOI:** 10.3390/molecules27144626

**Published:** 2022-07-20

**Authors:** Li-Yun Lin, Yi-Lun Liao, Min-Hung Chen, Shih-Feng Chang, Kuan-Chou Chen, Robert Y. Peng

**Affiliations:** 1Department of Food and Applied Technology, Hungkuang University, No. 1018, Sec. 6, Taiwan Boulevard, Shalu District, Taichung City 43302, Taiwan; lylin@hk.edu.tw (L.-Y.L.); sfchang@hk.edu.tw (S.-F.C.); 2Taiwan Seed Improvement and Propagation Station, COA. No. 6, Xingzhong St., Xinshe District, Taichung City 426017, Taiwan; liaoyl@tss.gov.tw; 3Agriculture & Food Agency Council of Agriculture, Executive Yuan, Marketing and Processing Division, Taichung City 43302, Taiwan; cmh@mail.afa.gov.tw; 4Graduate Institute of Clinical Medicine, College of Medicine, Taipei Medical University, No. 250, Wu-Xin St., Taipei 11031, Taiwan; robertpeng120@gmail.com; 5Department of Urology, Taipei Medical University Shuang-Ho Hospital, 250, Wu-Xin St., Xin-Yi District, Taipei 11031, Taiwan; 6School of Medicine and Nursing, Hungkuang University, No. 1018, Sec. 6, Taiwan Boulevard, Shalu District, Taichung City 43302, Taiwan

**Keywords:** coix, polyphenolics, antioxidant, anti-tyrosinase, facial masks

## Abstract

*Coix lacryma-jobi* var. *ma-yuen* L. Gramineae is widely cultivated in Taiwan. Literature regarding the molecular action mechanism of coixol on tyrosinase and the application of coicis seed extracts to the processing of facial masks is still lacking. Solvent extractability analysis revealed that most of the polyphenolics in coicis seeds were water soluble (3.17 ± 0.12 to 3.63 ± 0.07 μg/mLGAE). In contrast, the methanolic extract contained the most flavonoids (0.06 ± 0.00~0.26 ± 0.03 μg/mL QE) and coixol (11.43 ± 0.13~12.83 ± 0.14 μg/mL), showing potent antioxidant capability. Additionally, the contents of coixenolide (176.77 ± 5.91 to 238.60 ± 0.21 μg/g), phytosterol (52.45 ± 2.05 to 58.23 ± 1.14 mg/g), and polysaccharides (3.42 ± 0.10 to 4.41 ± 0.10 mg/g) were rather high. The aqueous extract (10 μg/mL) and the ethanolic extract (1 mg/mL) showed no cytotoxicity to B16F10 melanocytes. More attractively, the ethanolic extract at 1 mg/mL caused 48.4% inhibition of tyrosinase activity in B16F10 melanocytes, and 50.7% on human tyrosinase (hTyr) fragment 369–377. Conclusively, the coicis seed extracts containing abundant nutraceuticals with promising anti-hTyr activity and moisturizing capability can serve as good ingredients for facial mask processing.

## 1. Introduction

*Coix lacryma-jobi* L. var. *ma-yuen* Gramineae, widely planted in Taiwan, China, and Japan, is commonly called coicis (or coix), adlay, adlay millet, Job’s tears (USA, UK), Huanren, or Pearl barley (China). For many decades, it has served as a healthy food supplement in many Asian countries. Among all cultivars, *Coix lacryma-jobi* var. *ma-yuen* Stapf is mostly used, due to its soft shell [1]. Coicis seeds (Appendix A) contain abundant lipids, polysaccharides, lignans, polyphenols, and adenosines [2], and are widely utilized in traditional Chinese medicine for the treatment of various ailments, particularly cancer [3]. To date, a number of its biological activities have attracted a great deal of attention, with activities including antioxidant/free radical scavenging [4,5,6], anti-inflammatory [7,8], anti-mutagenic [9], anti-tumor, anticancer [10,11,12], hypolipidemics, hypocholesterolemics [6,13,14,15], hypoglycemics [13,16,17], and anti-obesity [18], etc. Coicis seeds consist of (g/100g) moisture 10.83, protein 13.05 (approximately double that of rice), fat 5.45, carbohydrate 68.60, fiber 0.36 and ash 1.3, and worth noting, coicis flour also contains 2.25% amylase [19,20].

Coicis seeds have also been reported to contain thiamine, riboflavin, niacin and ascorbic acid [21], as well as a diversity of active components, including coniferyl alcohol, syringic acid, ferulic acid, syringaresinol, 4-ketopinoresinol, mayuenolide, *p*-hydroxybenzaldehyde, vanillin, syringaldehyde, sinapaldehyde, coixol [4,22], trans-coniferylaldehyde and sinapaldehyde; coixenolide, caffeic acid, chlorogenic acid, coixspirolactams A-D, methyl dioxindole-3-acetate, ceramides, naringenin, gallic acid, and caffeic acid [4,23]. GC/MS analysis identified coix lactams and coixol to be present in the hull of coix seeds, and after polishing, the signals disappeared [22].

Recently, ‘Pearl and Job’s Tears Powder’ has been commercialized as a health food with the indications ‘skin whitening and moisturizing’ and ‘spleen strengthening and wet-repelling’. Some products are applied as facial masks, such as ‘Pearl Barley & Milk’ and ‘Clear Turn Princess Veil, Rich Moist Mask’, and so on.

The safest and most effective way to treat cutaneous hyperpigmentation is to reduce melanin production by inhibiting tyrosinase activity [24]. However, most tyrosinase inhibitors described in the literature lack clinical efficacy when incorporated into topical products [24]. Hydroquinone (IC_50_ > 4000 μM) and its derivative arbutin (IC_50_ > 4000 μM) only weakly inhibited hTyr (human tyrosinase), and kojic acid showed a weak efficacy (IC_50_ > 500 μM) [24]. Thus, it is apparent that the facial mask market requires a potent and reliable anti hTyr preparation. To promote the market values of coicis, we explored the molecular action mechanism of the coicis active components regarding its anti-tyrosinase bioactivity. On the other hand, considering the cost efficiency and overall bioactivity beneficial to the skin physiology, we accessed the common extraction technology to reclaim the active constituents from coicis seeds and screened the best cultivar for processing the facial masks.

## 2. Results and Discussion

### 2.1. The Comparison of Extractability by Different Solvent

The total extractable substance was found to be the highest by water extraction, reaching within a range 3.80 ± 0.09% (No. 2) to 4.55 ± 0.14% (No. 1), and the next was by ethyl acetate (Table 1). The ethanolic extractability was seen higher than that by methanolic (Table 1).

### 2.2. The Total Polyphenolic and Flavonoid Contents

The total polyphenolic content was also found to be the highest in aqueous extract (3.17 ± 0.12 μg/mL for No.1 to 3.63 ± 0.07 μg/mL for No. 5) (Table 2). The extractability by EtOH and MeOH was rather comparable, while ethyl acetate yielded the least (Table 2). The total flavonoid content present in the coix seeds was rather low compared to that of polyphenolics, and it seemed that the methanol brought forth to a higher extractability for flavonoids (Table 2). The main polyphenolics and flavonoids contained in coix seeds cited in the literature [4,23] are rearranged and shown in Appendix A. Some important heterocyclic compounds of coix, like coixspirolactams A–E, methyl dioxindole-3-acetate, and coixol (6-methoxy-2-benzoxazolinone), cited in the literature [4,23] are collected and listed in Appendix A.

### 2.3. Antioxidative Capability of Different Solvent Extracts from Coicis Seeds

The methanolic extract yielded the most potent antioxidative capability in view of the DPPH or the ABTS^+^ free radical scavenging capability (Table 3). The order for DPPH was (in μg/mL): No. 2 > No. 1 > No. 4. For ABTS^+^, the order was: No. 3 > No. 2 > No. 4 (Table 3). Thus, the antioxidant activity of coix seeds was more likely due to the total polyphenolic and flavonoid contents present in the methanolic extracts (Table 2 and Table 3). It is worth noting that higher extractability (Table 1) did not correlate with larger content of polyphenolic and flavonoids (Table 2; Appendix A); similarly, higher polyphenolic and flavonoid contents did not mean stronger antioxidant capability (Table 3). Recent literature has demonstrated that the number and location of phenolic hydroxyl of the flavonoids significantly influence the antioxidant activity [25]. Kinetically, the solubility of antioxidant constituents, the kinds of ROS species, and the molecular collision frequency in the reaction system may account for another reasons (Table 1, Table 2 and Table 3; Appendix A).

### 2.4. The Content of Phytosterols in the Coicis Seeds

No. 4 and No. 5 cultivars seemed to possess the highest amount of phytosterols (Table 4), the next were No. 2 > No. 1 > No. 3 (Table 4) (*p* < 0.05).

Phytosterols and ceramides (Appendix A) are effective in blocking the declining collagen synthesis after UV irradiation and even stimulating synthesis. They may be useful additions to anti-aging products [26]. Ceramides are important structural lipid components of the epidermis which pertinently serve as the important skin barrier function, playing a key role in maintaining homeostasis of the human body [27].

### 2.5. The Content of Crude Polysaccharides in the Coicis Seeds

Coicis water-soluble polysaccharide fractions are rich in water-soluble dietary fibers. The crude polysaccahride content ranged from 3.42 ± 0.10 mg/mL to 4.14 ± 0.10 mg/mL (Table 4) (*p* < 0.05). The total dietary fiber and soluble and insoluble dietary fiber contents (g/100 g, dry weight) reached 74.8, 71.9, and 2.9, respectively [14], while the water holding capacity reached 3.5 ± 0.1, 4.1 ± 0.2, and 3.2 ± 0.1 g/g, respectively, compared to 2.2 ± 0.1 g/g for the control (*p* < 0.05) [14], suggesting the feasibility of coicis extract for processing facial masks.

### 2.6. The Content of Coixol in Coicis Seeds

Methanolic extract exhibited the highest content of coixol (expressed in μg/mL), which was comparable among No. 1, No. 4, and No. 5. The next was those from aqueous extraction, which showed a range within 9.28 ± 0.20 μg/mL to 10.50 ± 0.12 μg/mL. The ethanolic extract contained a relatively low amount of coixol, and the ethyl acetate extract yielded the lowest content (Table 5).

Coixol is a natural product extracted from Coix Lachryma-Jobi var. ma-yuen var. ma-yuen Stapf [28]. Coicis semen refers to the dried ripe kernels of coix, which is rich in nutrients and compounds with various pharmacological activities [2], e.g., coixol appears to be responsible for antispasmodic actions [29]. Recent studies have shown that the active constituents in coicis semen could be used to treat flat wart, verruca vulgaris, and infectious condyloma [28], implicating a high potential of coix for processing of facial masks. Coixol inhibited the expression of MUC5AC mucin gene and production of MUC5AC mucin protein from NCI-H292 cells induced by EGF or TNF-α. Coixol decreased PMA-induced MUC5AC mucin secretion from NCI-H292 cells [30], beneficial to skin inflammation. Coixol exhibits certain anti-inflammatory bioactivities by inhibiting the expression of pro-inflammatory mediators in vitro [28]. The mechanism of action is related in part to its ability to inhibit the activation of NF-κB, MAPKs pathways, and NLRP3 inflammasome [28], which implies the protective effect against the skin inflammation.

In a βTC-6 cell model, coixol (200 μM) stimulated insulin secretion at high glucose concentration (20 mM) after being incubated for 60 min at 37 °C [31], which is beneficial for avoiding hyperglycemic pigmentation. Long-term type 2 diabetes with hyperglycemia, or high blood glucose tends to be associated with poor circulation, which reduces blood flow to the skin. Reduced blood circulation can lead to changes in the skin collagen profile and quality and, as a consequence, to changes in skin texture, appearance, and ability to heal.

### 2.7. The Content of Coixenolide in Coicis Seeds

The content of coixenolide in the coix seeds ranged from 176.77 ± 5.91 to 238.60 ± 0.21 μg/g (Table 5). Coixenolide, exhibiting antineoplastic activity, makes up no more than 0.25% of the coicis seeds [29].

Coixenolide is one component occurring in the coix seed oil [32]. Kanglaite (KLT), an aqueous microemulsion of an oil extracted from the Chinese crude drug coicis seeds [33], contains in majority coixenolide as its main constituent [4,33]. KLT is a new diphasic broad-spectrum antitumor drug against many types of tumor cells. The mechanism of action of coixenolide has been proposed to be via inhibition of the pathway involving NF-κB-dependent transcription [3,32,34].

### 2.8. Viability of B16F10 Melanocytes Affected by the Coicis Seed Extracts

The aqueous extracts of coicis seeds were more toxic than the ethanolic extracts. Suppression of the cell viability occurred at 0.5 mg/mL by aqueous extract (viability 88.91%), and further to 68.15% at 1 mg/mL (Figure 1a), compared to the positive kojic acid (93.32%) (Figure 1a). The IC_50_ was estimated to be 1.5 mg/mL for the aqueous extract (Figure 1a). By contrast, the cells were totally unaffected by the ethanolic extract at a dose 1 mg/mL, compared to 94.08% by kojic acid (Figure 1b). A slight inhibition of cell viability was seen to occur at doses ≥20 mg/mL, and the IC_50_ was estimated to occur at around 49 mg/mL (Figure 1b). Although a diversity of compounds, including hydroquinone, have been considered as tyrosinase inhibitors since the early 1990s [35], it is suggested that the specific cytotoxic properties of the target compounds are actually more important, not only for melanocytes, but also for their efficacy as inhibitors of melanogenesis [36,37,38].

### 2.9. The Inhibitory Effect on the Intracellular Tyrosinase and hTyrosinase Fragment 369–377

The commercially available mushroom isoenzymes AbPPO3 and AbPPO4 of tyrosinase (mTyr) have quite different amino acid sequences in the region of the active site, differing significantly from human tyrosinase (hTyr) [24]. hTyr exhibits unique binding sites with thiamidol at Ile368, Ser375 and Ser380 that are lacking in mTyr [24]. Such small changes in enzyime–ligand interactions would give rise to dramatic changes in binding affinities; as a consequence, the two diverse inhibition profiles from hTyr and mTyr in fact always show a large discrepancy [24], and similarly between the hTr and the mice B16F10 melanocytes. To date, only two categories of compound have been found to be effective for inhibiting hTyr, i.e., resorcinols and thujaplicins [39].

Compared to the negative control, the coicis aqueous extract at a dose ≤0.5 mg/mL did not reveal any inhibitory effect on either the intracellular tyrosinase or the hTyr (human tyrosinase fragment 369–377) (Table 6). A slight degree of inhibition occurred at 1.00 mg/mL of the aqueous extract, but the activity remained at 88.1 ± 1.6% and 91.0 ± 1.5%, respectively, for intracellular tyrosinase and the hTyr fragment 369–377 (Table 6). In contrast, the ethanolic extract exerted more potent inhibitory effect. At 1.0 mg/mL, the activity was suppressed to 51.6 ± 0.3% and 49.3 ± 0.8%, respectively (Table 6), implying the inferiority of the polyphenolics contained in the water extract of coix seeds with respect to the anti-tyrosinase activity when compared to the flavonoid content in the ethanolic extract (Table 2), which is consistent with Nguyen et al. (2016) [40]. Tyrosinase is the rate-limiting enzyme in melanin production; accordingly, it is the most prominent target for inhibiting hyperpigmentation [24].

Currently, it is generally acknowledged that the key regulatory enzyme of melanogenesis in melanocytes and melanoma cells is tyrosinase [41]. In summary, the process of melanogenesis can be roughly divided into three stages within a single biochemical pathway. The first stage involves the initiation of autooxidation of tyrosine catalyzed by tyrosinase (monophenolase) and/or with adjuvant peroxidase (i.e., hydroxylation of a monophenol) to produce L-DOPA, a reaction that requires dopa as a cofactor [42]. The second stage implicates the autooxidation of L-DOPA catalyzed by tyrosinase (diphenolase) and/or with adjuvant peroxidase) (the dehydrogenation of a catechol) to produce dopaquinone [43,44], and in the third stage, the peroxidase–H_2_O_2_ system plays the central role, acting solely or collaboratively with tyrosinase, to facilitate the oxidative polymerizations of pigment precursors and terminal stages of melanogenesis, or it may proceed spontaneously [42,45,46,47]. Suggestively, the strong anti-tyrosinase components contained in coix may play an important role at stages 1 and 2 by inhibiting the conversion of tyrosine into L-DOPA and then to dopaquinone. Apparently, the antioxidants could exert their contribution in the final stage of melanogenesis, and may be more or less present in stages 1 and 2 as well.

Zuo et al. (2018) [25] commented that the number and location of phenolic hydroxyl of the flavonoids significantly influence the anti-tyrosinase activities [25], which in fact is not consistent with many other findings. Recent literature has demonstrated that the most potent inhibitors of human tyrosinase were resorcinyl-thiazole derivatives, especially the newly identified Thiamidol (Beiersdorf AG, Hamburg, Germany) (isobutylamido-thiazolyl resorcinol), which had an IC_50_ of 1.1 μM, compared to 108 μM for mTyr [24]. Clinically, Thiamidol visibly reduced the appearance of age spots within 4 weeks, and after 12 weeks, some age spots were indistinguishable from the normal adjacent skin [24]. Additionally, omeprazole is a powerful anti-tyrosinase against the browning of apples [48]. Both Thiamidol and omeprazole are heterocyclic compounds. Thiamidol bears a thiazolyl nucleus; in contrast, omeprazole has a pyridino and a benzimidazole nucleus (see Appendix A), which may speculatively dock better with the active site of tyrosinase, resulting in more efficient inhibition. Speculatively, by comparing with the docking model reported by Mann et al. (2018) [24], coixol may be able to initiate its bindings at Ile368 (with thiazolyl), Phe347 (with thiazolyl), Val377 (with the benzoyl), and His202 (with the methoxy group of coixol), and Ser375 (with the carbonyl in benzoxazolinone) provided with slight distortion.

### 2.10. Tests on the Water Holding Capability of the Facial Masks

When tested on the moisture content, the commercialized mask samples responded differently, and the percent increase in moisturizing capability was found to range from 4.7% (Ref 1) to 11.6% (Ref 2). In contrast, the samples fabricated with the Coicis essence in this study showed values of 11.4% (CSE mask 1) and 10.2 % (CSE mask 2), respectively (Table 7). Obviously, such a result could be ascribed to four reasons: (1) the incorporation of hyaluronic acid, corn polyol and deep Moisturizer; (2) the presence of tremendous amounts of hydrophilic components in the aqueous extract (Table 1); (3) the huge content of water-soluble dietary fibers [14]; as well as (4) the crude polysaccharides present in the Coicis extract (Table 4). Coicis seed extract has the effect of whitening skin, and it can keep the human body skin glossy and exquisite, eliminating acne, senile plaque, and freckles, and improving rough skin (NHK, Health for Life). In addition, coicis seed powder can strengthen the immune system (NHK, Health for Life). Both the essence and the facial masks are easily accessible anytime and anywhere.

In summary, Figure 2 shows the overall whitening and water holding mechanism of the coix seed components. Coixol inhibited tyrosinase, while the polyphenolics and flavonoids inhibited the bioactivity of peroxidase, which was suggested by Mastore et al. (2005) [44] to be involved in the melanin formation; as a consequence, the transformation from tyrosine to L-DOPA and then to DOPAquinone was inhibited, otherwise the DOPAquinone could be converted to melanin via oxidative polymerization, while the phytosterols and polysaccharides exerted strong water holding bioactivity on the epidermis to furnish a moisturizing and whitening capability.

## 3. Materials and Methods

### 3.1. Source of Coix Seeds

Coicis seeds (Taichung No. 1~Taichung No. 5) (Appendix A) were gifted by the local gross sales at Nan-Tou County, Taiwan. The seeds were harvested from the year 2019 to 2020 in batch wise collections. Each batch weighed 1 kg. A total of 8 batches were collected. The seeds were kept at 4–15 °C during transportation and rinsed quickly in laboratory with double distilled water twice for 2 min, immediately followed by tissue-wiping and blow drying. The desiccated fresh samples were stored at −20 °C until use.

### 3.2. Chemicals and Reagents

2,2-diphenyl-1-picrylhydrazyl (DPPH *) (0.5 mM in methanol), 2,2-azinobis-(3-ethylbenzothiazoline-6-sulfonic acid) (ABTS^+^), and ethylenediaminetetraacetic acid (EDTA) were provided by E. Merck (Dresden, Germany). Reference standard coixol (6-methoxy-2-benzoxazolinone; 6-MBOA) (purity 98.21%) was supplied by Med Chem Express (MCE, South Brunswick Township, NJ, USA). Coixenolide (CAS No. 29066-43-1) [(2S,3R)-3-[(Z)-hexadec-9-enoyl]oxybutan-2-yl] (*E*)-octadec-11-enoate was purchased from Neostar United Industrial Co., Ltd. (Binjiang Economic Development Zone, Changzhou, Jiangsu 213033, China) (www.neostarunited.com, accessed on 11 June 2022). The human tyrosinase fragment 369–377 was provided by Sigma Aldrich.

### 3.3. Sources of Cell Line

The B16F10 cells (ATCC CRL-6475, BCRC60031) were obtained from the Bioresource Collection and Research Center (BCRC, Hsinchu, Taiwan). The cells were maintained in DMEM (Hyclone, Logan, UT, USA) supplemented with 10% fetal bovine serum and 1% antibiotics at 37 °C, 5% CO_2_ in a humidified incubator.

### 3.4. Solvent Extractability

To 10 g of pulverized coicis seed powder, 200 mL of deionized water (or ethanol 95%, or methanol, or ethyl acetate) was added, and the mixture was maintained at 60 °C for 3 h with constant stirring. The extract was decanted to recover the solvent extract. The extractions were repeated for three times, the respective extracts were combined and subjected to rotary evaporation under reduced pressure until 100 mL to obtain the extracts of water or ethanol 95%, or methanol, or ethyl acetate, respectively.

### 3.5. Determination of Total Phenolic Acids and Flavonoids

The determination of the total phenolics and flavonoids were carried out by following the method as previously reported in our laboratory [49].

### 3.6. DPPH Free Radical Scavenging Capability

The measurement of the DPPH radical scavenging activity was performed according to Brand-Williams et al. (1995) [50]. Briefly, to 0.5 mL sample extracts, 0.3 mL DPPH radical reagent and 3 mL absolute ethanol were added. The optical density (changes in color from deep violet to light yellow) were monitored with a UV/VIS spectrophotometer (DU 800; Beckman Coulter, Fullerton, CA, USA) at 517 nm for 100 min against a blank containing a mixture of ethanol (3.5 mL) and DPPH radical reagent (0.3 mL). The scavenging activity percentage (*SA*%) was calculated from Equation (1).
(1)$$SA\%=100-[100\times (A_{sample}-A_{blank})/Acontrol]$$

### 3.7. ABTS^+^ Antioxidant Capability

ABTS^+^ Antioxidant Assay Kit (Cat# AOX-1) (ZenBios, West Bengal, Indian) was used to evaluate the antioxidant capability of different extracts of coicis seeds by following the instructions given by the manufacturer.

### 3.8. Determination of the Phytosterol Content

The phytosterols were determined according to the method as previously reported [49].

### 3.9. Extraction of Water-Soluble Polysaccharides

Chen et al. was followed with slight modification [51]. In brief, the desiccated sample coix seeds were pulverized and defatted in a preparative Soxhlet apparatus (RBS company, Taoyuan, Taiwan). To the defatted residue, a 20-fold volume of deionized water was added, sonicated with 10 KW, refluxed at 95 °C for 4 h, and filtered. The extraction was repeated three times. The following step was carried out as directed. The purified polysaccharides were combined and lyophilized for use.

### 3.10. Isolation and Purification of Coixol

The protocol described by Li and Liu (2011) [52] was followed to prepare high-purity coixol. The coixol obtained was then further purified in our laboratory. In brief, desiccated coix seeds were pulverized and sieved through #80 mesh. Then, 1000 g coix powder was percolated with 20 L sulfuric acid solution (1%). The following procedure was performed as cited. The coixol obtained was then further recrystallized in acetone–petroleum ether. Physicochemically, coixol is colorless and appears as needle like crystals (acetone–petroleum ether), m.p. 159–160 °C.

### 3.11. Isolation and Purification of Coixenolide

Ukita and Tanimura (1961) [53] was followed with slight modification to carry out the isolation procedures. In brief, desiccated coix seeds were pulverized and sieved through #80 mesh. The powder (1000 g) was accurately weighed and subjected to preparative Soxhlet extraction with acetone (2.5 L) for 3 h. The syrup was dissolved in 250 mL petroleum ether, filtered, and the filtrate was concentrated in the rotary evaporator to yield reddish brown syrup. The syrup was dissolved in 50 mL petroleum ether, applied to a silica gel column (60 × 450 mm) and eluted with 800 mL petroleum ether. The eluate was evaporated in a rotary evaporator to give residue 1 (R1). R1 was redissolved in 50 mL petroleum ether, shaken twice with 40 mL 0.2 N KOH solution. The aqueous KOH layer was discarded. The alkali insoluble neutral substance (NS) was evaporated in a rotary evaporator to give residue RN1. The following procedures were carried out as directed using the alumina and silica gel (ASG) column for further separation. The chloroform eluate containing the NS was evaporated to give residue RN2. To advance the purity, 250 mg of the RN2 was subjected to silica gel column chromatographic separation (stationary phase: silica gel, column 10 mm × 350 mm), and sequentially eluted with petroleum ether containing ether (4%, 10%, and 50% ether in petroleum ether). The fractions eluted by 50:50 ether in petroleum were collected, combined, filtered, and evaporated under reduced pressure to obtain pure coixenolide ($n_{D}^{20}$1.4702–1.4705, $\alpha_{D}^{20}$0^o^). The purified coixenolide was used as the reference standard in the following assay for colixenolide content in coix seeds.

### 3.12. Determination of Coixol

The content of coixol in the extracts of coicis seeds was determined according to Ali et al. (2017) [54]. In brief, the HPLC analyses of the standard coixol and coicis extracts were performed on Agilent 1200 SL Rapid Resolution HPLC-UV system (Agilent Technologies, Singapore). An amount of 2.0 μL of sample solution was injected. The mobile phase consisted of two solutions. Solution A (0.1% TFA in water, *v/v*) and solution B (methanol) were operated in a linear gradient manner: 15% B from 0–1 min, 15–50% B from 1–2 min, 50–95% B from 2–6 min, maintained at 95% 6–7 min, and 95–15% B from 7–8 min. The column temperature was maintained at 30 °C [54].

### 3.13. Assay of Coixenolide

The method of Yang et al. (2004) [55] was employed with modification. Briefly, the desiccated pulverized (#80) coix seed powder was extracted in a preparative Soxhlet extractor with petroleum ether. Crude oil (1 g) obtained was dissolved in 20 mL of 7% (*w/w*) methanolic HCl solution, subjected to acid-catalyzed transesterification by refluxing in a water bath at 100 °C for 4 h cooled and neutralized with 30% methanolic sodium methoxide (Sigma). The next steps were carried out as cited. The coixenolide content in the coix seeds was calculated from the calibration curve against 2,3-butanediol content obtained from different amount of coixenolide standards obtained in the above.

### 3.14. Cell Viability Affected by Different Coicis Seed Extracts

The B16F10 cells (1 × 10^5^ cells/mL) were evenly pipetted into 10 culture tubes, 5 mL in each. In tubes 1–5, the aqueous extract was tested at 0 to 1.0 mg/mL; the ethanolic extract was tested in tubes 6–10 at 0 to 50 mg/mL. Kojic acid (1 mM) was used as the positive control. The tubes were further incubated in a humidified incubator at 37 °C under an atmosphere of 5% CO_2_ for 24 h. The absorbance of the broth was read at 600 nm using a microplate reader Gen5™ (BIO-TEK Instrument, Vermont, VT, USA). The percent viability was calculated taking the absorbance of the control as 100%.

### 3.15. Assay for the Intracellular Tyrosinase Activity

According to Yang et al. (2006) [56], the cells obtained from the above were disrupted by ultrasonication, centrifuged at 3000× *g* to discard the cell debris. The supernatant cell extract (100 μL) was mixed with freshly prepared L-DOPA solution (0.1% in phosphate-buffered saline) and incubated at 37 °C. The absorbance of this solution at 490 nm was measured with a microplate reader Gen5™ (BIO-TEK Instrument, Vermont, VT, USA) to estimate the production of dopachrome. Corrections were made for auto-oxidation of L-DOPA.

### 3.16. Assay for the Anti-Human Tyrosinase Fragment 369–377 Activity

The isolated coixol 100 mg was dissolved in 100 mL 95% ethanol to serve the stock solution (1 mg/mL). The inhibition assay for human tyrosinase fragment 369–377 (hTyr 369–377) was conducted as follows. In brief, different amount of coix seed extracts (aqueous extract 0.01, 0.50, and 1.00 mg/mL; ethanolic extracts 1.0, 20.0, 50.0 mg/mL), coixol (5, 10, 15 μg/mL), and 5 mM DOPA (in 50 mM sodium phosphate buffer, pH 6.8) were transferred into 96-well microtiter plate. To each well, hTyr 369–377 (200 units in 10 μL) was added. The reaction mixture was incubated at 37 °C for 30 min. The amount of dopachrome produced in the reaction mixture was determined spectrophotometrically at 490 nm (OD_490_) using the ELISA microplate reader. Experiments were performed triplicate, and the inhibition percentage was evaluated using Equation (2):(2)$$\%\;Inhibition_{tyrosinase\;activity}=\;100\times (B-A)/A$$
where *B* is the OD_490_ values of the blank control, and *A* is the OD_490_ values for the coix seed extract treated group. Corrections were made for auto-oxidation of L-DOPA.

### 3.17. Preparation of the Facial Masks

The recipes for preparation of the coicis seed essence consisted of hyaluronic acid 3%, corn polyol 3%, deep moisturer 3%, deionized water 70.3%, antioxidant 0.1%, hydrophilic antibacterial agent 0.6%, and coicis seed extract 20%. In brief, the deionized water was heated to 50 °C, to which the antioxidant was added in portions. The mixture was cooled to ambient temperature and all other ingredients were added, mixed well to obtain the coicis seed essence. For preparation of the masks, the mask films, made of either bi-axially oriented polypropylene (BOPP) or cast polypropylene (CPP), were dipped into the coicis essence for 20 min to fully soak up the essence. The face masks were then removed and sealed in the packaging vinyl bags.

### 3.18. Test for Moisture Holding Capability

Skin moisture analyzer (Amazon, Co., Marston Gate, UK) was used to measure the percent skin moisture (IRB No: HK-HSSI-005 approved by HungKuang University, Taichung, Tiwan).

### 3.19. Statistical Analysis

The experiment was performed in triplicate for each substance. The results were expressed in percentage with respect to control values and compared by one-way ANOVA and Tukey’s test. A difference was considered statistically significant when *p* ≤ 0.05.

## 4. Conclusions

Facial skin is very delicate and vulnerable to environmental damage, including from chemical, physical, mechanical, and biological agents. Chemical damage may originate from UV screeners, whiting agents, moisturizers, emulsifies and face cleansing agents. Physical damage may involve radiation, thermal injuries, and improper sonication. Mechanical damage can be caused by underqualified massagers. In addition, biological damage could be induced by microbial infections coming from unsanitary or expired cosmetics. The development of facial masks from natural sources could avoid most of the above-mentioned sources of damage. Coicis seeds bear several nutraceutical and biomedical advantages, involving the promising anti-hTyr components coixol and others, as well as high contents of antioxidant and antimicrobial polyphenolics, flavonoids, and terpenes, high moisture holding and anti-inflammatory polysaccharides, phytosterol and ceramics. By carefully integrating the formulation and processing, coicis seeds are extremely feasible materials for facial mask fabrication.

## Figures and Tables

**Figure 1 molecules-27-04626-f001:**
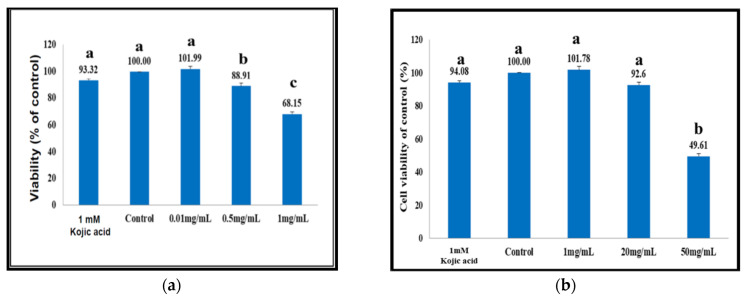
The viability test on B16F10 melanocytes affected by the different coicis extracts. (**a**) Aqueous extracts. (**b**) Ethanolic extracts. The superscripts in lower case (a, b and c) in the figure indicate significant difference between the columns (*p* < 0.05).

**Figure 2 molecules-27-04626-f002:**
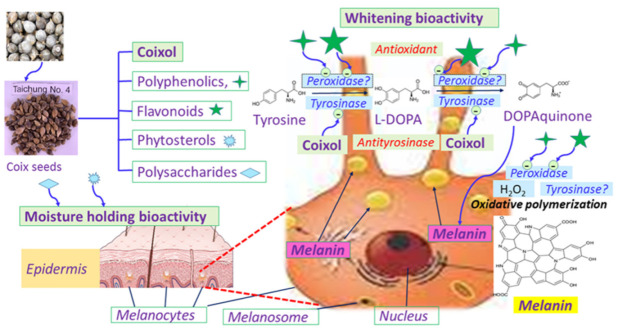
Mechanism of action of different coix constituents on epidermis to exert whitening and water holding bioactivity. Coixol inhibited tyrosinase to prevent the transformation of tyrosine to L-DOPA and then to DOPAquinone, which is converted to melanin via oxidative polymerization. The polyphenolics and flavonoids inhibited the bioactivity of peroxidase, involved in the melanin formation, while the phytosterols and polysaccharides exerted the water holding bioactivity on the epidermis. Involvement of peroxidase suggested by Mastore et al. [44].

**Table 1 molecules-27-04626-t001:** Comparison of the solvent extractability from the coix seeds *.

	Solvent	Extraction Yield, %			
Cultivar		Water	EtOH	MeOH	Ethyl Acetate
No. 1		^A^ 4.55 ± 0.14 ^a^	^B^ 1.75 ± 0.04 ^c^	^D^ 1.11 ± 0.03 ^d^	^C^ 2.69 ± 0.12 b ^b^
No. 2		^C^ 3.80 ± 0.09 ^a^	^C^ 1.61 ± 0.05 ^cd^	^B^ 1.73 ± 0.13 ^c^	^A^ 3.21 ± 0.06 ^b^
No. 3		^B^ 4.19 ± 0.13 ^a^	^C^ 1.65 ± 0.02 ^d^	^A^ 2.01 ± 0.04 ^c^	^C^ 2.65 ± 0.07 ^b^
No. 4		^B^ 4.26 ± 0.12 ^a^	^B^ 1.89 ± 0.01 ^c^	^C^ 1.34 ± 0.15 ^d^	^B^ 3.02 ± 0.03 ^b^
No. 5		^B^ 4.13 ± 0.11 ^a^	^A^ 2.05 ± 0.06 ^c^	^DE^ 1.07 ± 0.04 ^d^	^AB^ 3.13 ± 0.01 ^b^

* Data are expressed as mean ± SD from triplicate experiments (*n* = 3). The superscript in upper case indicates significantly different in the same column. The superscript in lower case indicates significantly different in the same row (*p* < 0.05).

**Table 2 molecules-27-04626-t002:** The solvent extractability for total polyphenolics and total flavonoids from the coix seeds *.

Extraction	Total Polyphenolics (Gallic Acid eq., μg/mL)			
Cultivar	Water	EtOH	MeOH	Ethyl Acetate
No.1	^CD^ 3.17 ± 0.12 ^a^	^E^ 0.60 ± 0.02 ^c^	^B^ 1.10 ± 0.03 ^b^	^C^ 0.38 ± 0.00 ^d^
No.2	^AB^ 3.58 ± 0.11 ^a^	^C^ 1.01 ± 0.05 ^b^	^D^ 0.68 ± 0.05 ^c^	^B^ 0.51 ± 0.03 ^d^
No. 3	^C^ 3.27 ± 0.03 ^a^	^A^ 1.36 ± 0.10 ^b^	^D^ 0.73 ± 0.09 ^c^	^D^ 0.28 ± 0.01 ^d^
No. 4	^B^ 3.44 ± 0.20 ^a^	^D^ 0.91 ± 0.01 ^c^	^A^ 1.48 ± 0.10 ^b^	^A^ 0.62 ± 0.05 ^d^
No. 5	^A^ 3.63 ± 0.07 ^a^	^B^ 1.18 ± 0.02 ^b^	^C^ 0.98 ± 0.07 ^bc^	^C^ 0.37 ± 0.00 ^d^
Total flavonoids (Quercetin eq., μg/mL)				
No. 1	^CD^ 0.01 ± 0.00 ^c^	^BC^ 0.05 ± 0.01 ^b^	^A^ 0.26 ± 0.03 a	^A^ 0.05 ± 0.00 ^b^
No. 2	^CD^ 0.01 ± 0.00 ^d^	^A^ 0.11 ± 0.00 ^a^	^C^ 0.09 ± 0.01 ^ab^	^A^ 0.05 ± 0.01 ^c^
No. 3	^B^ 0.04 ± 0.00 ^b^	^BC^ 0.05 ± 0.00 ^ab^	^D^ 0.06 ± 0.00 ^a^	^A^ 0.06 ± 0.02 ^a^
No. 4	^C^ 0.02 ± 0.00 ^c^	^B^ 0.06 ± 0.01 ^b^	^B^ 0.17 ± 0.02 ^a^	^BC^ 0.02 ± 0.00 ^c^
No. 5	^A^ 0.06 ± 0.00 ^b^	^BC^ 0.05 ± 0.01 ^bc^	^A^ 0.20 ± 0.03 ^a^	^B^ 0.03 ± 0.00 ^d^

* Data are expressed as mean ± SD from triplicate experiments (*n* = 3). The superscripts in upper case indicate significantly different in the same column. The superscripts in lower case indicate significantly different in the same row (*p* < 0.05).

**Table 3 molecules-27-04626-t003:** The free radical scavenging capability for DPPH and ABTS^+^ by different extracts of the coix seeds *.

Extraction	DPPH (Trolox µg/mL)			
Cultivar	Water	EtOH	MeOH	Ethyl Acetate
No. 1	^A^ 1.96 ± 0.02 ^c^	^AB^ 3.55 ± 0.01 ^b^	^AB^ 8.29 ± 0.10 ^a^	^A^ 1.05 ± 0.03 ^d^
No. 2	^D^ 1.57 ± 0.02 ^c^	^C^ 3.25 ± 0.03 ^b^	^A^ 8.52 ± 0.06 ^a^	^A^ 1.16 ± 0.04 ^d^
No. 3	^A^ 1.93 ± 0.04 ^c^	^A^ 3.64 ± 0.07 ^b^	^C^ 7.90 ± 0.00 ^a^	^B^ 0.93 ± 0.01 ^d^
No. 4	^BC^ 1.78 ± 0.01 ^c^	^A^ 3.67 ± 0.01 ^b^	^B^ 8.04 ± 0.00 ^a^	^C^ 0.45 ± 0.00 ^d^
No 5	^B^ 1.84 ± 0.03 ^c^	^D^ 3.19 ± 0.00 ^b^	^D^ 7.67 ± 0.03 ^a^	^D^ 0.36 ± 0.03 ^d^
**ABTS^+^ (Trolox µg/mL)**				
No. 1	^B^ 2.50 ± 0.01 ^c^	^C^ 2.74 ± 0.02 ^b^	^E^ 6.45 ± 0.11 ^a^	^A^ 2.16 ± 0.04 ^d^
No. 2	^A^ 2.62 ± 0.06 ^b^	^D^ 2.42 ± 0.06 ^c^	^B^ 8.04 ± 0.08 ^a^	^CD^ 1.77 ± 0.14 ^d^
No. 3	^AB^ 2.56 ± 0.00 ^c^	^B^ 3.16 ± 0.04 ^b^	^A^ 8.79 ± 0.12 ^a^	^C^ 1.87 ± 0.09 ^d^
No. 4	^C^ 2.20 ± 0.02 ^c^	^A^ 3.50 ± 0.00 ^b^	^C^ 7.46 ± 0.01 ^a^	^AB^ 2.09 ± 0.01 ^cd^
No. 5	^D^ 2.03 ± 0.02 ^c^	^A^ 3.49 ± 0.00 ^b^	^D^ 6.86 ± 0.06 ^a^	^E^ 1.60 ± 0.03 ^d^

* Data are expressed as mean ± SD from triplicate experiments (*n* = 3). The superscripts in upper case indicate significantly different in the same column. The superscripts in lower case indicate significantly different in the same row (*p* < 0.05).

**Table 4 molecules-27-04626-t004:** The phytosterol and polysaccharide contents of the coix seeds *.

Cultivar	Phytosterol (mg/g)		
	Campesterol	γ−Σιτοστερολ	Total
No. 1	^B^ 1.71 ± 0.20	^B^ 52.20 ± 2.03	^C^ 53.91 ± 1.95
No. 2	^B^ 1.79 ± 1.00	^B^ 51.63 ± 3.63	^B^ 53.42 ± 3.00
No. 3	^AB^ 1.82 ± 0.10	^BC^ 50.63 ± 2.63	^C^ 52.45 ± 2.05
No. 4	^A^ 1.95 ± 0.22	^A^ 59.92 ± 3.65	^A^ 61.87 ± 4.17
No. 5	^A^ 1.91 ± 0.14	^A^ 56.32 ± 1.20	^AB^ 58.23 ± 1.14
**Crude Polysaccharides (mg/mL)**			
No. 1	^A^ 4.14 ± 0.10		
No. 2	^D^ 3.42 ± 0.10		
No. 3	^B^ 3.63 ± 0.14		
No. 4	^B^ 3.69 ± 0.20		
No. 5	^C^ 3.59 ± 0.02		

* Data are expressed as mean ± SD from triplicate experiments (*n* = 3). The superscripts in upper case indicate significantly different in the same column (*p* < 0.05).

**Table 5 molecules-27-04626-t005:** Solvent extractability for coixol and the recovery of coixenolide from coix seeds *.

Cultivar	Recovery of Coixol (μg/mL)			
	Water	EtOH	MeOH	Ethyl Acetate
No. 1	^B^ 9.97 ± 0.10 ^b^	^A^ 5.59 ± 0.01 ^c^	^A^ 12.83 ± 0.14 ^a^	^A^ 1.24 ± 0.00 ^d^
No. 2	^CD^ 9.28 ± 0.20 ^b^	^C^ 5.19 ± 0.00 ^c^	^C^ 11.43 ± 0.13 ^a^	^C^ 1.04 ± 0.03 ^d^
No. 3	^C^ 9.69 ± 0.03 ^b^	^C^ 5.11 ± 0.01 ^c^	^AB^ 11.92 ± 0.10 ^a^	^A^ 1.23 ± 0.04 ^d^
No. 4	^A^ 10.50 ± 0.12 ^b^	^B^ 5.43 ± 0.01 ^c^	^A^ 12.01 ± 0.15 a	^AB^ 1.19 ± 0.12 ^d^
No. 5	^A^ 10.40 ± 0.02 ^b^	^A^ 5.53 ± 0.03 ^c^	^A^ 12.18 ± 0.11 ^a^	^A^ 1.29 ± 0.02 ^d^
**Recovery of Coixenolide (µg/g)**				
No. 1	^CD^ 176.77 ± 5.91			
No. 2	^C^ 184.16 ± 4.30			
No. 3	^B^ 203.26 ± 4.21			
No. 4	^A^ 234.19 ± 4.42			
No. 5	^A^ 238.60 ± 0.21			

* Data are expressed as mean ± SD from triplicate experiments (*n* = 3). The superscripts in upper case indicate significantly different in the same column. The superscripts in lower case indicate significantly different in the same row (*p* < 0.05).

**Table 6 molecules-27-04626-t006:** The anti-tyrosinase activity in *B16-F10 melanocytes* and hTyrosinase fragment 369–377 affected by different treatments.

Sample	Dose	Tyrosinase Activity *, %	Activity of hTyr **, %
Control	-	100.0 ± 0.0	100.0 ± 0.0
α-MSH, μg/mL	1.67	117.8 ± 1.0	-
Kojic acid, mg/mL	0.28	54.4 ± 0.1	88.7 ± 1.5
Coixol, μg/mL	5	49.4 ± 0.6	35.7 ± 0.4
	10	38.5 ± 0.2	24.2 ± 0.6
	15	29.4 ± 0.6	13.8 ± 0.5
Aqueous extract, mg/mL	0.01	104.3 ± 0.2	100.5 ± 0.4
	0.50	101.6 ± 3.0	101.8 ± 2.4
	1.00	88.1 ± 1.6	91.0 ± 1.5
Ethanolic extract, mg/mL	1.0	51.6 ± 0.3	49.3 ± 0.8
	20.0	46.1 ± 0.9	43.5 ± 0.6
	50.0	34.3 ± 0.7	32.4 ± 0.4

* The activity of intracellular tyrosinase was normalized to the cell numbers. ** hTyr: human tyrosinase fragment 369–377.

**Table 7 molecules-27-04626-t007:** The test on the water holding capability of facial masks.

Sample Masks ^a^	Water Holding Capability, %		% Enhanced
	Before Treatment	After Treatment	
Ref 1	38.9	43.6	4.7
Ref 2	42.2	53.8	11.6
CSE 1	37.4	48.8	11.4
CSE 2	39.4	49.6	10.2

^a^ Ref 1 (applied on the left arm) and Ref 2 (applied on the right arm): reference masks purchased from commercial brands. CSE: coix seed extract facial masks. CSE 1 (applied on the left arm); CSE 2 (applied on the right arm).

## Data Availability

The data presented in this study are available in Appendix A.

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
