# Peer review of "Molecular Action Mechanism of Coixol from Soft-Shelled Adlay on Tyrosinase: The Future of Cosmetics"

_molecules, 2022, doi:10.3390/molecules27144626_

Round 1
Author Response
Dear Reviewer,
The manuscript has been revised and we submit the "Responses to reviewers" file for your evaluation.
Thank you
Dr. Kuan Chou Chen

Reviewer 2 Report
The manuscript «Anti-Human Tyrosinase Activity of Coixol with Flavonoids Together with Phytosterols and Polysaccharides Implicate Feasibility for Whitening and Moisturizing Bioactivity» was reviewed.
General impression: the manuscript is not designed according to the journal requirements, the section "results and discussion" needs to be redone. Part of the chemical composition is poorly described. What method is used to determine the qualitative composition of phenolic compounds? what kind of sample? nothing is clear at all. The division of substances into groups is not clear.
In its current form, I do not recommend publication, the manuscript requires major revision. Special notes below:
I recommend changing the title to a more concise one, for example:
Coixol and bioactive compounds from Coix lacryma-jobi var. ma-yuen (or Coix lacryma-jobi) from Taiwan: potential anti-human tyrosinase activity
or
Molecular action mechanism of coixol from Soft-Shelled Adlay (or Job’s tears) on tyrosinase: the future of cosmetics.
according to the requirements of the Molecules journal, the sections "materials and methods" and the section "Results and discussions" should be swapped. Highlights are not needed. Please follow the Figure and Table requirements. The list of references is not designed according to the requirements.
The place of growth and harvesting of Coicis seeds is not exactly clear. Specify. What year are the seeds? storage conditions?
Why “Taichung No.1- 5” and Reference sample “f” were so different?
change “Authentic” to "reference standard"
The extraction process and also points 2.3-2.8 is not exactly described. Check all conditions. And how much and what kind of extracts did you get?
Points 2.9-2.14 should be reduced.
Results. Do not describe the data of the tables in the text, give an analysis and conclusions.
There is no supporting material, although the authors refer to it all the time.
For chemical formulas using a professional editor, apply the same style everywhere.
The pharmacology part is hopeful that the manuscript has novelty, but the text should also be revised.
Author Response

(The authors gave the same response as above.)

Round 2
Reviewer 1 Report
Accept
Reviewer 2 Report
The authors have made all the corrections. There are no more comments.